# The Impact of the COVID-19 Pandemic on Binge Eating Disorder: A Systematic Review

**DOI:** 10.3390/nu15173777

**Published:** 2023-08-29

**Authors:** Alice Caldiroli, Davide La Tegola, Francesca Manzo, Alberto Scalia, Letizia Maria Affaticati, Enrico Capuzzi, Fabrizia Colmegna, Marios Argyrides, Constantinos Giaginis, Leonardo Mendolicchio, Massimiliano Buoli, Massimo Clerici, Antonios Dakanalis

**Affiliations:** 1Department of Mental Health, Fondazione IRCSS San Gerardo dei Tintori, Via G.B. Pergolesi 33, 20900 Monza, Italy; alice.caldiroli@irccs-sangerardo.it (A.C.); davide.lategola@irccs-sangerardo.it (D.L.T.); e.capuzzi1@campus.unimib.it (E.C.); fabrizia.colmegna@irccs-sangerardo.it (F.C.); massimo.clerici@unimib.it (M.C.); 2Department of Medicine and Surgery, University of Milano Bicocca, Via Cadore 38, 20900 Monza, Italy; f.manzo5@campus.unimib.it (F.M.); a.scalia5@campus.unimib.it (A.S.); letizia.affaticati@gmail.com (L.M.A.); 3Department of Psychology, Neapolis University Pafos, Paphos 8042, Cyprus; m.argyrides.1@nup.ac.cy; 4Department of Food Science and Nutrition, School of Environment, University of Aegean, 81400 Myrina, Greece; cgiaginis@aegean.gr; 5Istituto Auxologico Italiano IRCCS, U.O. dei Disturbi del Comportamento Alimentare, Ospedale San Giuseppe, 28824 Piancavallo, Italy; l.mendolicchio@auxologico.it; 6Department of Pathophysiology and Transplantation, University of Milan, Via Festa del Perdono 7, 20122 Milan, Italy; massimiliano.buoli@unimi.it; 7Department of Neurosciences and Mental Health, Fondazione IRCCS Ca’ Granda Ospedale Maggiore Policlinico, Via F. Sforza 35, 20122 Milan, Italy

**Keywords:** binge eating disorder, COVID-19 pandemic, coronavirus, lockdown, eating disorders

## Abstract

The aim of this systematic review was to synthesise the impact of the COVID-19 pandemic on binge eating disorder (BED) the new onset and course. Inclusion criteria: original articles and BED diagnosis; and the main outcomes: relationships between the COVID-19 pandemic and the new onset/clinical changes in BED, and specific results for BED. Exclusion criteria: mixed/inaccurate diagnoses and articles not written in English. We searched four databases and one registry until 5 May 2023. The quality appraisal was conducted using the Effective Public Health Practice Project (EPHPP) tool. Twelve studies with 4326 participants were included. All studies were observational with nine cross-sectional and three longitudinal. Four of the included studies investigated new-onset BED, while eight examined the BED clinical course of patients with a previous diagnosis. With the exception of one study, the available literature indicates both an increase in BED diagnoses and a clinical worsening during COVID-19. Major limitations include study quality (weak-to-moderate) and high heterogeneity in terms of pandemic phase, population, geographical areas, and psychometric tools. Our findings indicate that BED patients are particularly vulnerable to events characterised by social distancing and over-worry, and should be, therefore, carefully monitored. Further studies are needed to corroborate our findings, implement preventive strategies, and promote personalised treatments. PROSPERO registration number: CRD42023434106

## 1. Introduction

The worldwide spread of SARS-Cov-2 in December 2019 led the World Health Organization (WHO) to declare that the coronavirus disease (COVID-19) was a pandemic on 11 March 2020 [1]. After the introduction of restrictive measures to reduce the infection (e.g., lockdown), a number of subjects experienced a worsening of mental well-being as a result of fear of contagion and social distancing [2,3].

Several authors reported critical impacts of the COVID-19 pandemic on eating disorders (EDs). In particular, three systematic reviews [4,5,6] and one meta-analysis [7] revealed an increased prevalence or re-exacerbation of EDs with an increase in related hospitalisations during lockdowns. Nevertheless, differences between the different diagnostic entities were rarely explored, and contrasting results on anorexia nervosa (AN) and bulimia nervosa (BN) were described. Some studies reported more binge eating (BE) episodes after lockdowns in patients affected by BN [8], while other authors failed to identify a significant impact of the pandemic on the body mass index (BMI) of subjects suffering from AN and BN [6,7]. Curiously, the effects of the COVID-19 pandemic on individuals diagnosed with binge eating disorder (BED) have been poorly explored till now, as more attention has been given to trans-diagnostic BE attitudes.

A BED is defined by the presence of objective BE episodes concomitantly with a sense of lack of control and without inappropriate compensatory behaviours [9]. Although the 11th revision of the International Classification of Diseases (ICD-11) [10] included the subjective binges in BED diagnostic criteria, it has been demonstrated that there is an agreement between ICD-11 and DSM-5 for the same ED diagnosis in 98% of the cases [11]. Of note, BED represents the most common ED, with a lifetime prevalence of 3.5% among women and 2% among men [12,13]. In females under 30 years of age, the prevalence varies between 0.6% and 6.1%, according to the country of origin [14].

A specific BE behaviour, as well as BE episodes, is also characteristic of other clinical conditions such as BN, emotional eating, or night eating syndrome. Nevertheless, in the 5th edition of the Diagnostic and Statistical Manual of Mental Disorders (DSM-5), BED was recognised as a specific condition, finally acquiring its own nosographic dignity [9]. In fact, BED is characterised by specific clinical features such as emotion dysregulation, impulsivity, and cognitive distortion, and underpinned by peculiar neurobiological mechanisms [15]. Moreover, BED is often concomitant with medical (e.g., metabolic syndrome) and psychiatric comorbidities, which together worsen the BED course and impair the quality of life of the subjects suffering from this condition [16,17].

At the same time, the persistence of BED symptoms can be enhanced by social and environmental factors [18]. For example, family-related adverse life experiences (ALEs), especially early loss of caregivers, family separations, and detrimental parental interactions, negatively impact BN and BED patients [19]. Moreover, Forester and colleagues [20] demonstrated that specific times of the day (in particular, dinner time, but also lunch time and late evening) favour binge eating in BED patients, being the periods with the strongest food cravings and difficulties in emotional regulation.

Focusing on the pandemic period, some authors identified risk/protective factors for the new onset of BE symptoms (not BED diagnosis) [21], including both European and United States (US) data [22]. For example, having a mental disorder in comorbidity, the presence of pandemic-associated worry, self-reported pandemic-related loneliness, and being female and belonging to an ethnic minority were all identified as risk factors for the onset or exacerbation of BED [21,22]. In contrast, suffering from the COVID-19 illness, a change in living situation, being a key worker, or being in treatment emerged as protective factors against BE onset/worsening during the restrictions [21].

The COVID-19 pandemic was a dramatic event and, especially the confined, represented a challenging environment for individuals with BED. To date, it remains unclear whether BED psychopathology benefitted from lockdown measures or became worse, and evidence regarding the impacts of the COVID-19 pandemic on patients with the BED is still scarce. In light of these considerations, the objective of the present review was to systematically summarise the current literature about the impacts of the COVID-19 pandemic on the new onset and course of the BED.

## 2. Methods

### 2.1. Search Strategy

In order to find relevant articles, a search in the main databases (PubMed, Web of Science, PsychInfo), as well as consultation in US registries (National Institutes of Health, ClinicalTrials.gov), was performed. The search was conducted from 1987 to 5 May 2023, using the keywords “((COVID-19) OR (SARS-Cov-2) OR (coronavirus) OR (lockdown)) AND ((binge eating) OR (binge eating disorder))”. No filters or limits were used.

### 2.2. Eligibility Criteria

Inclusion criteria were (1) original articles; (2) reported diagnosis of BED; (3) when the main topic of the article focused on the relationship between the COVID-19 pandemic and new onset or clinical changes in the BED; and (4) when different diagnoses were considered only papers reporting results on the BED subgroup were taken into account. Exclusion criteria were (1) reviews, meta-analyses, commentaries, letters, case reports, pooled analyses, comments, case studies, and study protocols; (2) when there were mixed or not accurately described diagnoses (e.g., BE behaviour); or (3) when articles were not written in the English language.

### 2.3. Study Selection 

Two authors searched and screened the titles and abstracts of each record independently. Potentially eligible records were flagged, and the full text was obtained. Full texts were then reviewed for final inclusion. A third author supervised the process and resolved doubts and disagreements.

### 2.4. Data Extraction

Two authors extracted relevant data independently; doubts and disagreements were resolved by a third author. The following information was extracted: paper author and title, publication year, time period during which the study was conducted, geographic area where the research was conducted, study design, population, inclusion and exclusion criteria, sample size, mean age, data on BMI, psychometric tools, and main findings. If relevant data were not reported in the selected articles, the corresponding author was contacted to obtain further information.

### 2.5. Quality Appraisal

Two reviewers assessed the quality of each included study using the “Quality Assessment Tool for Quantitative Studies” developed by the Effective Public Health Practice Project (EPHPP) [23]. The tool evaluates the following domains: selection bias, study design, confounders, blinding, data collection method, withdrawals, and dropouts. Each domain is rated as “strong”, “moderate” or “weak”. Global quality is then calculated as strong (no weak ratings), moderate (one weak rating), or weak (two or more weak ratings). A third author supervised the process and resolved any discrepancies.

### 2.6. Data Synthesis

Relevant data were tabulated. The following information was presented: author/publication year, period of interest, geographic area where the research was conducted, study design, population, inclusion and exclusion criteria, sample size, mean age, data on BMI, psychometric tools, main findings, and quality score. We also provided a narrative synthesis of the studies’ characteristics and main findings.

This systematic review, including the search strategy and inclusion/exclusion criteria, followed the Preferred Reporting Items for Systematic Reviews and Meta-Analyses (PRISMA) guidelines [24,25]. The protocol was pre-registered on PROSPERO (CRD42023434106).

## 3. Results

### 3.1. Study Selection

The search of all the databases provided a total of 6728 citations. Among these, 12 were identified as duplicates. The number of records marked as ineligible by the automation tools was 6582. Of the 134 records screened, 32 studies were discarded because, after reviewing the abstracts, the papers dealt with another topic. Reviews and meta-analyses (n = 18), as well as case reports, case series, surveys, letters and commentaries (n = 4), were excluded. The full texts of the remaining 80 citations were examined in more detail, concluding that 68 studies did not meet the inclusion criteria because 38 did not provide specific results on the BED, and 30 studies analysed subjects without an accurate diagnosis of the BED. Twelve studies met the inclusion criteria and were included in the present review (Figure 1).

### 3.2. Study Characteristics

The characteristics of each study were summarised in Table 1, Table 2 and Table 3. A total of 4326 participants from 12 studies were included in the present review. All studies were observational. Nine had a cross-sectional design [26,27,28,29,30,31,32,33,34], two had a longitudinal retrospective design [35,36], and one had a longitudinal prospective design [37]. The mean ages ranged from 19.3 to 45.5 years. Most studies were conducted in the US, Brazil, or in a European country. Four studies investigated the new onset of BED during the COVID-19 pandemic. Eight studies assessed pandemic-driven changes in the clinical course of patients with a previous diagnosis of the BED.

### 3.3. Quality Appraisal

The quality appraisal was conducted using the “Quality Assessment Tool for Quantitative Studies” developed by the EPHPP [23]. Two-thirds of the included studies were rated as “moderate” in quality [26,27,28,29,34,35,36,37] (Table 1 and Table 2); four studies were rated as “weak” [30,31,32,33] (Table 3). None of the studies received a “strong” quality rating.

### 3.4. BED New Onset during COVID-19 Pandemic

We included four studies about the new onset of the BED during the COVID-19 pandemic. Among them, two were performed on the Brazilian population [26,27] and two on the US population [28,37].

Pimentel and his group [26] published a prospective, cross-sectional study assessing the relationship between the COVID-19 pandemic and the occurrence of BED and mental suffering in a sample of Brazilian health professionals. The online interview was completed by 219 professionals, of which 35 (16%) reported BED-related symptoms. This study also found a higher prevalence of BED in women than men and a direct association between BED onset and high BMI. Similarly, a cross-sectional and retrospective study was carried out by Garcês and colleagues [27], who compared 94 (60.1%) normal-weight adults and 129 (39.9%) overweight/obese subjects. The results of this study confirmed the increased risk of developing BED during social distancing both in the overweight/obese group and in normal-weight adults.

The negative impact of the COVID-19 pandemic on eating behaviour was also demonstrated by Bronfman and Chao [28] who examined, differently from the first two studies, a population of US college students between February and April 2021. In this cross-sectional study, factors potentially associated with ED symptoms were analysed, finding that individuals with more severe depressive symptoms had higher odds of meeting criteria for BN, BED, and subthreshold EDs, while more severe anxiety symptoms and more frequent social media use were associated with increased odds of BED. 

Finally, an interesting longitudinal study by Murray and collaborators [37] was conducted to update annual incidence rates and demographic trends of EDs among military service members between 2017 and 2021. Their results demonstrated that the annual incidence rates of all EDs increased from 2017 to 2019, decreased in 2020, and increased again in 2021. From the first year of the surveillance period (2017) to the endpoint (2020), the BED annual incidence rate tripled.

All included studies investigating the relationship between the COVID-19 pandemic and BED new onset are detailed in Table 1.

### 3.5. BED Clinical Changes/Relapse during COVID-19 Pandemic

Eight studies were included about the impacts of the COVID-19 pandemic on BED clinical courses. Most authors considered the COVID-19 pandemic and related lockdowns as a traumatic event, which promoted dysfunctional eating behaviours.

Four studies that analysed the first lockdown period in different countries reported similar results. In the study by Termorshuizen et al. [29], 1021 subjects (216 with a self-reported diagnosis of BED) were recruited from the US and the Netherlands (NL) via social media. Lifetime diagnosis of a BED represented 35% of the US sample and 23% of the Dutch one. Both patients affected by BED and BN reported more frequent BE episodes, especially of stockpiled foods. In the second study, Giel et al. [35] investigated ED symptoms, as well as general psychopathology, in 42 patients with a history of a BED who previously sought treatment in the randomised–controlled IMPULS trial in Germany. The authors assessed self-reported ED pathology using the Eating Disorder Examination (EDE) and demonstrated an increase in BE episodes in the four weeks during lockdown compared to the four weeks before lockdown. The cross-sectional study by Freizinger et al. [33] analysed a sample of 22 adolescents and young adults affected by BED. The authors used the binge eating scale (BES), which is a self-administered questionnaire assessing eating behaviours, feelings, and cognitive aspects linked to BE, and the COVID-19 Exposure and Family Impact Survey-Adolescent and Young Adult version (CEFIS-AYA), which investigates experiences related to the COVID-19 pandemic. They found that the degree of COVID-related stress was associated with more BE behaviours. More recently, Baenas and colleagues [34] evaluated clinical changes in patients with EDs during the first lockdown in a sample of 81 female patients. Clinical information was collected using the COVID Isolation Eating Scale (CIES), a self-report questionnaire composed of four subscales investigating the circumstances of the quarantine, the effects of the lockdown on eating symptoms, the effects of the confinement on eating behaviours, and the impacts of remote interventions. The authors found that post-lockdown BED patients experienced weight gain and a worsening of anxiety, depression and emotional dysregulation. These findings confirmed previous research by the same group that had found similar results in a multicentre study using the same instrument (CIES) but focused on the second wave of the COVID-19 pandemic. A larger sample (N = 829) was recruited, including 113 individuals with a BED diagnosis, who reported the highest impact on weight and eating style impairment in comparison with other subgroups and an increase in anxiety–depressive symptoms [36].

De Pasquale and collaborators [31] investigated, in a sample of 469 Italian college students, the relationship between mood states and eating behaviours during the first year of the COVID-19 pandemic. The utilised online protocol included the following psychometric tools: the Fear of COVID-19 Scale (FCV-19S) for the evaluation of fear, the Profile of Mood States (POMS) for the assessment of emotional states, the Eating Disorder Inventory-2 (EDI-2) and the BES. A significant effect of fear of COVID-19 in worsening BE behaviours was identified by the authors.

On the other hand, Colleluori et al. [30], although analysing a large sample of BED patients (N = 174) during the same period (first lockdown), reported different findings. The data were retrieved from an online survey, investigating the experience of healthcare providers (HCPs) involved in the treatment of EDs. The objective of this study was to assess whether social restrictions represented a stressor for ED patients and their HCPs, if they affected therapeutic alliance and altered the frequency of dysfunctional behaviours. The authors found a high variability in the changes in frequency of BE episodes after the pandemic, which increased in 30.5% and decreased in 24.7% of BED subjects. Similarly, Frayn et al. [32] observed both symptom deterioration and improvement in the BED, analysing seven BED patients. Participants underwent a semi-structured interview and authors identified factors that may decrease BE, such as having greater control over the food environment.

A summary of the studies on the BED clinical course during COVID-19 is reported in Table 2 and Table 3.

## 4. Discussion

Most studies evaluating the impacts of the COVID-19 pandemic on EDs, focused on unspecific BE behaviours or BE episodes that are symptoms shared by the different EDs and characterise other clinical conditions, such as obesity or emotional eating. BE episodes are core symptoms of BED [9,10], which represents a defined nosographic entity underpinned by specific neurobiological mechanisms [38,39]. To the best of our knowledge, this is the first systematic review that synthesises the impact of the COVID-19 pandemic and the related sanitary actions on the BED, diagnosed according to DSM-5 [9]. Focusing on the specific disorder, rather than unspecific binge eating symptoms, may shed light on some pathophysiological mechanisms underlying the BED and its etiopathogenesis, and may foster the implementation of targeted preventive/treatment strategies for subjects suffering from this specific ED.

Two main findings emerged from our systematic literature review:the prevalence of the BED during the COVID-19 pandemic increased based on data collected from Brazil and the US;the patients affected by the BED experienced a worsening of symptoms during the first year of the COVID-19 pandemic.


The four studies investigating the impacts of pandemic lockdowns on new diagnoses of BED found an increase in the prevalence of BED and its symptoms [26,27,28,37]. Interestingly, a direct association between the BED and BMI levels during the first pandemic wave was identified [26,27], as well as a higher prevalence of the BED in female than male HPs [26], in line with previous reports [40].

Six papers demonstrated a clinical worsening of the BED during the first lockdown as shown by more BE episodes [29,35], the elevation of the BMI, and more severe anxiety and depressive symptoms, as well as emotion dysregulation and impaired eating styles [34,36]. These findings confirm the available data in the literature [6,7,41,42], however, focused on unspecific BE behaviours regardless of BED diagnosis. In contrast, the paper by Machado et al. [43] found no significant change in symptoms and BMI after the lockdown confinement in a sample of patients suffering from EDs; however, only two patients had a diagnosis of BED, thus supporting the hypothesis that this group of subjects might suffer more from the negative consequences of social distancing. In addition, the two weak-quality studies outlined mixed results about the impacts of the pandemic on the clinical course of the BED [30,32]. It must be specified that the Italian study by Colleluori and colleagues [30] collected indirect information with online surveys addressed to HCPs.

The studies described in this review suggest that the higher frequency of the BED or the worsening of its course may be mediated by other factors influenced by the pandemic, such as the presence of depression/anxiety [28,34,36], fear of being infected [31], or an increase in BMI [26,34,36]. Of note, after the outbreak of the pandemic, BMI increased in the general population, especially among women and the youngest [44] which represent groups vulnerable to developing BED [45]. Similarly, symptoms such as depression and anxiety have also been frequently reported in the general population since the outbreak of the pandemic [46]. On the other hand, the presence of affective symptoms in patients affected by EDs may have potentiated the negative effects of social isolation and distancing on the clinical course of these individuals [47]. These observations suggest a role of neuropeptides, such as oxytocin, a molecule implicated in social behaviour [48] and appetite regulation [49,50], in explaining the complex entanglement of depressive symptoms, worsening of BED symptoms, weight gain, and social distancing. In support of this hypothesis, previous research suggested that the administration of intranasal oxytocin is able to reduce food intake and cravings [51]. 

### Limitations

This systematic review presented different limitations. First of all, the relatively small sample size of the included studies, as well as the cross-sectional design of most papers, with self-reported clinical information, may have contributed to making the quality of studies predominantly moderate or weak. Second, the high heterogeneity of the included papers in terms of pandemic phase, population, geographical areas, and psychometric tools may reduce the power of the reported findings. The various phases of the pandemic were characterised by important differences in terms of the magnitude of restrictions, social distancing, and the use of telemedicine [52]. Moreover, the studies included in the present review were almost all conducted in Europe and America. Only one study [36] included a group of patients from Asia. Of note, the epidemiology of EDs differs across countries, being that eating behaviours are strongly influenced by the environment [53,54] and cultural factors, such as body image models proposed by the media [55]. In particular, the American continent has the highest prevalence for all EDs, followed by Asia and Europe, while data coming from Africa are scarce [56]. Finally, it should be noted that our results referred to very heterogeneous samples, including HPs, college students, and military service members, as well as the general population, limiting the generalizability of the results presented in this review.

## 5. Conclusions

In conclusion, our systematic review confirmed that social restrictions related to the COVID-19 pandemic have negatively impacted EDs, especially the BED. The strong entanglement between the restrictions, an elevated BMI, and the presence of depressive symptoms could have promoted the development or the worsening of the BED. The included studies presented some methodological weaknesses, making it difficult to draw definitive conclusions. Further research is necessary to clarify the underlying clinical and biological factors triggering the worsening of the BED when dramatic events occur, such as a pandemic. The final purpose is to foster more personalised interventions in order to prevent and promptly treat BED symptoms in predisposed and vulnerable subjects.

## Figures and Tables

**Figure 1 nutrients-15-03777-f001:**
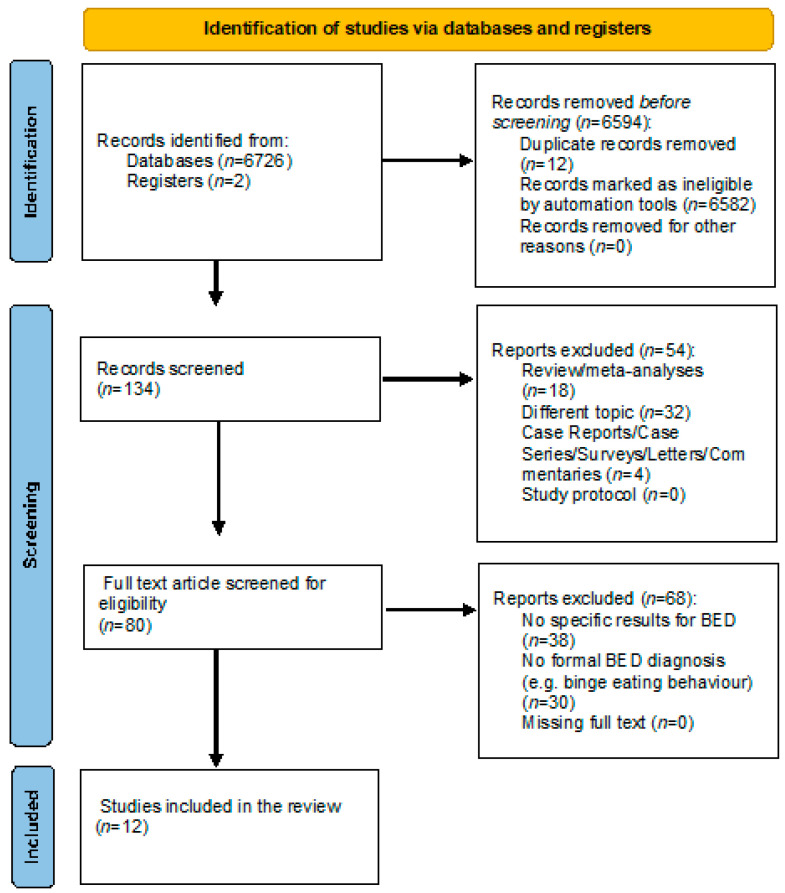
PRISMA diagram for systematic reviews.

**Table 1 nutrients-15-03777-t001:** Summary of findings about onset of Binge Eating Disorder during COVID-19 pandemic.

Study	Study Period	Country	Study Design	Data Collection Method/Psychometric Tools	Population	IC, EC	Sample Size (Females; Males; Non-Binary)	Mean Age (±SD)	BMI (at the Time of Study Initiation)	Main Findings	Quality Score ^1^
Pimentel et al., 2021 [26]	June 2020 – October 2020	Brazil	Cross-sectional	Online questionnaire: EDS-7	HPs	IC: age 18–60; worker in a Brazilian hospital involved in the fight against COVID-19 EC: diagnosis of ED	219 (183; 36; 0)	35.6 (±13.6)	Underweight: 6 (2.7%) Normal weight: 104 (47.5%) Overweight: 71 (32.4%) Obesity grade I: 29 (13.2%) Obesity grade II: 8 (3.7%) Obesity grade III: 1 (0.5%)	- BED-related symptoms: 35 (16%) - BED: F (85.7%) > M - Association between BED and BMI (*p* = 0.03)	Moderate (weak data collection methods)
Garcês et al., 2022 [27]	23 October 2020 – 3 December 2020	Brazil	Cross-sectional, retrospective	Online questionnaires: BES and IPAQ Answers related to 2 times points: (1) pre-pandemic period; (2) during social distancing, i.e., 23 October 2020–03 December 2020	General population	IC: age 18–60; agreeing to the digital ICF). EC: refusal to agree to the ICF; children, adolescents, and elderly; incomplete answers and/or non-completion of questionnaires.	323 (218; 105; 0)	30.24	Normal weight: 194 (60.1%) Overweight or obese: 129 (39.9%)	- ↑ BE predicted body mass gain (*p* < 0.001) during social distancing - ↑ prevalence of BED both in the normal weight and in the overweight/obese group during social distancing (*p* < 0.001)	Moderate (weak data collection methods)
Bronfman and Chao, 2023 [28]	February 2021 – April 2021	US	Cross-sectional	Online questionnaire: EDE-Q, EPII, GAD-7, PHQ-8, SMUI and questions based on the YRBSQ	College students	IC: college students in the US; age 18–23; completion of depression and EDs questionnaires. EC: N/A.	202 (168; 30; 4)	20.7 (±1.7)	Mean BMI: 23.2 (±4.2)	- BED: 27.2% - ↑ depressive symptoms = ↑ odds of BED (*p* < 0.001) - ↑ anxiety symptoms = ↑ odds of BED (*p* < 0.001) - ↑ social media use = ↑ odds of BED (*p* = 0.03)	Moderate (weak selection bias)
Murray et al., 2023 [37]	1 January 2017 – 31 December 2020	US	Longitudinal prospective	DMSS; PHA: MHA questions	Military Members (in service)	IC: active component service members from all military services from 1 January 2017 to 31 December 2021. Answer “yes” to questions on “major life stressors”, “history of mental health care”, “PTSD screening” and “depression screening” EC: hospitalisation or visits for an ED diagnosis prior to the surveillance period	2454 ED diagnoses (1556; 898; 0)	N/A	N/A	-BED annual incidence rate = 0.7 per 10.000 p-yrs -From the first year of the surveillance period to the endpoint, BED annual incidence rates tripled	Moderate (weak selection bias)

Legend: ^1^ according to [23]; ↑ = increased; BE = binge-eating; BED = binge eating disorder; BEDS-7 = 7-item binge eating disorder screener; BES = binge eating scale; BMI = body mass index; DMSS = defence medical surveillance system; EC = exclusion criteria; EDs = eating disorders; EDE-Q = Eating Disorders Examination Questionnaire; EPII = epidemic-pandemic impact inventory; F = females; GAD-7 = generalised anxiety disorder-7 scale; HPs = health professionals; IC = inclusion criteria; ICF = informed consent form; IPAQ = short version of the International Physical Activity Questionnaire; M = males; MHA = mental health assessment; *p* = *p* value; N/A = not available; PHA = periodic health assessment; PHQ-8 = patient health questionnaire; p-yrs = per years; PTSD = post-traumatic stress disorder; SD = standard deviation; SMUI = social media usage index; US = United States; YRBSQ = Youth Risk Behaviour Survey Questionnaire.

**Table 2 nutrients-15-03777-t002:** Summary of findings from moderate-quality studies on BED clinical changes/relapse during COVID-19 pandemic.

Study	Study Period	Country	Study Design	Data Collection Method/Psychometric Tools	Population	IC, EC	Sample Size (Females; Males; Other)	Mean Age (±SD)	BMI (at the Time of Study Initiation)	Main Findings	Quality Score ^1^
Termorshuizen et al., 2020 [29]	April 2020 – May 2020	NL and US	Cross-sectional	Online survey: quantitative measures and free-text responses investigating the impact of COVID-19 on eating disorders, general physical and mental well-being (GAD-7), treatment of EDs	Individuals with self-reported ED	IC: minimum age 18 in US, minimum age 26 in NL; life-time self-reported ED EC: N/A	1021 (1001; 18; 2) 216 affected by BED	US: 30.61 (±9.37) NL: N/A	N/A	- ↑ BE episodes and urges to binge in BN and BED - BE of stockpiled food: BED and BN > AN (*p* < 0.001)	Moderate (weak data collection methods)
Giel et al., 2021 [35]	Spring 2020	Germany	Longitudinal retrospective Comparison between 4 weeks immediately before lockdown (on a recall basis) and 4 weeks during lockdown	BE episodes: EDE Other assessments: BDI-II, ERQ, PSQ, SCID, SOC-L9	BED patients (who had previously participated in the IMPULS trial, investigating a novel psychotherapy for BED)	IC: BED diagnosis according to DSM-5; minimum age 18 years. EC: suicidality, pregnancy or lactation, a severe mental or somatic disorder	42 (34; 8; 0)	45.5 (±12.6)	Mean BMI: 34.0 (±6.9)	↑ BE episodes in the 4 weeks during lockdown vs 4 weeks before lockdown (*p* < 0.001)	Moderate (weak confounder)
Baenas et al., 2022 [36]	August 2020 – January 2021	Europe (Austria, Czech Republic, Germany, Lithuania, Portugal, Russia, Spain, Ukraine) and Asia (China, Korea, Japan)	Longitudinal retrospective Comparison pre- vs. post- lockdown	CIES	EDs patients	IC: ED diagnosis according to DSM-5 criteria, performed by expert clinical psychologists and psychiatrists using a semi-structured clinical interview (SCID-5). EC: N/A	829 (584; 245; 0) 113 affected by BED	27.9 (±12.3).	BED Mean BMI: 35.08 (±9.63)	- Patients affected by BED experienced the highest impact on weight and clinical symptoms in comparison with other EDs during lockdown. -BED patients during post- vs pre-lockdown: --↑ weight (*p* < 0.001; Cohen’s d = 0.12) -- ↑ BMI (*p* < 0.001; Cohen’s d = 0.14) -- impaired eating style (*p* = 0.01; Cohen’s d = 0.18) -- ↑ anxiety-depression symptoms (*p* = 0.004; Cohen’s d = 0.16)	Moderate (weak confounder)
Baenas et al., 2023 [34]	June 2020 – March 2021	Brazil, Portugal, Spain	Cross-sectional (retrospectively collected data) Comparison post- vs pre- lockdown	CIES (subscales I, II, III)	EDs patients	IC: females; ED diagnosis according to DSM-5 EC: N/A	264 (264; 0; 0) 81 affected by BED	33.49 (±12.54)	BED Mean BMI: 33.70 (±7.37)	BED patients during post- vs pre-lockdown: - ↑ weight (*p* < 0.001; Cohen’s d = 0.18) - ↑ BMI (*p* < 0.001; Cohen’s d = 0.20) - ↑ anxiety-depression (*p* < 0.001; Cohen’s d = 0.73) - ↑ emotional dysregulation (*p* < 0.001; Cohen’s d = 0.28)	Moderate (weak selection bias)

Legend: ^1^ according to [23]; ↑ = increased; AN = anorexia nervosa; BDI-II = Beck Depression Inventory-II; BE = binge-eating; BED = binge eating disorder; BMI = body mass index; BN = bulimia nervosa; CIES = COVID Isolation Eating Scale; DSM = Diagnostic and Statistical Manual of Mental Disorders; EC = exclusion criteria; EDE = eating disorder examination; EDs = eating disorders; ERQ = Emotion Regulation Questionnaire; F = females; GAD-7 = Generalised Anxiety Disorder-7 scale; IC = inclusion criteria; M = males; NL = Netherlands; p = p value; PSQ = Perceived Stress Questionnaire; SCID = Structured Clinical Interview for DSM-5; SOC-L9 = Sense of Coherence Scale-Leipzig 9 items; US = United States; vs = versus.

**Table 3 nutrients-15-03777-t003:** Summary of findings from weak-quality studies on BED clinical changes/relapse during COVID-19 pandemic.

Study	Study Period	Country	Study Design	Data Collection Method/Psychometric Tools	Population	IC, EC	Sample Size (Females; Males; Other)	Mean Age (±SD)	BMI (at the Time of Study Initiation)	Main Findings	Quality Score ^1^
Colleluori et al., 2021 [30]	9 March 2020 – 8 May 2020 (first lockdown)	Italy	Cross-sectional	Ad hoc online survey investigating the experience of HCPs in the management of EDs (AN, BN, BED) during phase I lockdown	Patients with DSM-5-defined EDs (AN, BN, BED)	IC: DSM-5-defined AN, BN, BED EC:_other ED (e.g., night eating syndrome, vigorexia or ortorexia)	453 (N/A) 174 affected by BED	N/A	N/A	- BE frequency: increased in 30.5% and reduced in 24.7% of patients affected by BED - ↑ BE episodes BN > BED (*p* < 0.05) - switching ED category: BN > BED or AN (*p* = 0.001) - ↑ visits frequency: AN = BN = BED (*p* = 0.113)	Weak (weak selection bias and weak data collection method)
De Pasquale et al., 2021 [31]	March 2020 – February 2021	Italy	Cross-sectional	Online questionnaires: BES, EDI-2, FCV-19s, POMS	College students	N/A	469 (248; 221; 0)	22.47 (±2.70)	N/A	- Fear of COVID-19 → ↑ BES scores (*p* < 0.01) - BED: F > M (*p* = 0.03)	Weak (weak selection bias and weak data collection methods)
Frayn et al., 2021 [32]	August 2020	US	Cross-sectional	Individual, semi-structured interview (conducted by a trained researcher) investigating: - perceived impact of COVID-19 on ED symptoms prior to starting treatment - experience with teletherapy - feedback on adapting treatment to address COVID-related concerns	Patients with BE spectrum disorders (undergoing a therapy treatment program for BE spectrum disorders, i.e., COMPASS)	IC: age 18–70; at least 12 objective or subjective binge episodes in the past 3 months; meeting DSM-5 criteria for an ED EC: BMI < 16; already receiving ED treatment; requiring immediate treatment for medical complications resulting from ED symptoms; severe psychopathology that would limit their ability to comply (e.g., severe depression with suicidal intent, active psychotic disorder)	11 (7; 3; 1) 7 affected by BED	42.8 (±14.2)	Mean BMI: 34.7 (±10.3)	Qualitative findings: -both symptom worsening and improvement during COVID-19 -Social distancing and stay-at-home measures were found to both improve and worsen symptoms for different patients	Weak (weak selection bias and weak data collection methods)
Freizinger et al., 2022 [33]	November 2020 – January 2021	Northeastern US	Cross-sectional (retrospectively collected data: questions pertained to the initial lockdown period of the COVID-19 pandemic)	Online survey: BES, CEFIS-AYA	Patients referred for evaluation and treatment for BED, emotional eating, or lifestyle counselling for weight concerns and/or preparation for bariatric surgery	IC: age 13–26; evaluated and/or treated from March 2019 to March 2020. EC: N/A	39 (31; 6; 2) 22 affected by BED	19.3 (±3.2)	N/A	- ↑ COVID-related stress = ↑ BES scores (*p* = 0.04) - ↓ age = ↑ BES scores (*p* = 0.02) - No association between BES scores and: -- food availability at home (*p* = 0.97) -- worriedness about having sufficient food in the home (*p* = 0.40) -- difficulty obtaining food (*p* = 0.84)	Weak (weak selection bias and data collecting method)

Legend: ^1^ according to [23]; ↑ = increased; ↓ = decreased; AN = anorexia nervosa; BE = binge-eating; BED = binge eating disorder; BES = binge eating scale; BMI = body mass index; BN = bulimia nervosa; CEFIS-AYA= COVID-19 Exposure and Family Impact Survey-Adolescent and Young Adult Version; DSM = Diagnostic and Statistical Manual of Mental Disorders; EC = exclusion criteria; EDI-2 = eating disorder inventory-2; EDs = eating disorders; F = females; FCV-19S = fear of COVID-19 scale; HCPs = healthcare providers; IC = inclusion criteria; M = males; p = p value; POMS = profile of mood states; US = United States.

## Data Availability

The datasets generated during and/or analysed during the current study are available from the corresponding author on reasonable request.

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
