# Peer review of "The Impact of the COVID-19 Pandemic on Binge Eating Disorder: A Systematic Review"

_nutrients, 2023, doi:10.3390/nu15173777_

Round 1
Reviewer 1 Report
This systematic review on the impact of the Corona pandemic on BED is timely and well written and was conducted following the PRISMA statement.
I have a few minor suggestions / questions for a revision:
The INTRODUCTION outlines the topic comprehensively and makes reference to other work in the field.
The authors only make reference to DSM-5, but should also mention BED in the novel ICD-11 classification.
The authors summarize that the pandemic was a “traumatic event”, however, I think this statement is quite bold and an overgeneralization.
General remark: I suggest to replace “Psychiatric disorder” by “mental disorder”.
METHODS
The databases named are not “psychiatric databases” but larger ones for all sorts of subjects / research fields.
The authors write that they performed the review according to PRISMA which is a strength.
Please clarify a few sentences: e.g. “at least two authors performed the selection of studies” – what does this mean? Were there actually three authors selecting? Please clarify. Please clarify use of automated tools. Were the raters independent? How large was agreement between raters?
Was the review pre-registered?
For the inclusion criteria: were there rules for how the BED diagnosis had to be assessed e.g. by clinicians or an interview etc.?
The RESULT section is a bit long and there is redundancy of text with results presented in tables.
Table 1 – please add author name and year in the first column.
Please carefully check for spelling and grammar – e.g. “ has showed” instead of “shown”.
Please carefully check for spelling and grammar – e.g. “ has showed” instead of “shown”.
Author Response
First of all, we would like to thank the First Reviewer for the interest in our manuscript and the useful suggestions. We revised the manuscript according to your indications.
I have a few minor suggestions / questions for a revision:
The INTRODUCTION outlines the topic comprehensively and makes reference to other work in the field.
The authors only make reference to DSM-5, but should also mention BED in the novel ICD-11 classification.
Thank you for the suggestion. We mentioned the ICD-11 classification in the Introduction.
The authors summarize that the pandemic was a “traumatic event”, however, I think this statement is quite bold and an overgeneralization.
Thank you for the observation. We replaced the word “traumatic” in the Introduction and Conclusions sections with the word “dramatic”, which sounds more cautious.
General remark: I suggest to replace “Psychiatric disorder” by “mental disorder”.
Thank you for the suggestion. We replaced the words.
METHODS
The databases named are not “psychiatric databases” but larger ones for all sorts of subjects / research fields.
Thank you for the observation. We removed the word “psychiatric”.
The authors write that they performed the review according to PRISMA which is a strength.
Thank you for the appreciation.
Please clarify a few sentences: e.g. “at least two authors performed the selection of studies” – what does this mean? Were there actually three authors selecting? Please clarify.
Thank you for the notice, the sentence sounds ambiguous. More precisely, two authors performed the selection of studies (F.M. and A.S.). We clarified this aspect in the manuscript, extensively revising the methods section according to the PRISMA checklist.
Please clarify use of automated tools. Were the raters independent? How large was agreement between raters?
Thank you for the questions. Some automated tools are available to support systematic searches (Khalil et al., 2022. PMID: 34896236 DOI: 10.1016/j.jclinepi.2021.12.005; Oliveira Dos Santos et al., 2023 PMID: 37187321 DOI: 10.1016/j.jbi.2023.104389). They are used in the very early stage of the search. The raters were independent, with an agreement of 95% of the records screened. A third author supervised the process and resolved doubts and disagreements. We specified this aspect in the Methods section.
Was the review pre-registered?
Yes, the review was pre-registered and we added the PROSPERO registration number in the Methods section.
For the inclusion criteria: were there rules for how the BED diagnosis had to be assessed e.g. by clinicians or an interview etc.?
Thank you for the request. No, there were no rules to assess BED diagnosis.
The RESULT section is a bit long and there is redundancy of text with results presented in tables.
Thank you for the observation. We simplified the results section in order to reduce the redundancy with tables.
Table 1 – please add author name and year in the first column.
Thank you, we added author name and year of the publication in the first column as you suggested for more clarity.
Please carefully check for spelling and grammar – e.g. “ has showed” instead of “shown”.
Thank you for the suggestion, we carefully check English spelling and grammar.
Reviewer 2 Report
The manuscript entitled „ The Impact of the COVID-19 Pandemic on Binge Eating Disorder: a Systematic Review” presents interesting issue, but some problems should be corrected.
Major:
(1) The main problem with the presented study is the fact that Authors declare that their systematic review and meta-analysis “followed the Preferred Reporting Items for Systematic Reviews and Meta-Analyses (PRISMA) guidelines” (lines 112-114), but in fact the manuscript is not prepared according to the recommendations of PRISMA, which are very specific and should be rigorously followed. Authors should get familiar with PRISMA checklist (http://prisma-statement.org/prismastatement/Checklist.aspx) and they should correct their manuscript to be prepared according to the checklist. E.g., the Abstract should include: background, objectives, data sources, study eligibility criteria, participants, and interventions, study appraisal and synthesis methods, results, limitations, conclusions and implications of key findings, systematic review registration number, while a number of elements is not presented in the Abstract of the submitted manuscript. However, Authors should correct the whole study (not only the Abstract Section).
(2) Especially while analysing the Materials and methods Section, a majority of required information is not presented, so in fact the majority of essential information is not provided, so the quality of this systematic review can not be assessed.
(3) Authors conducted the systematic review based on the searching from 2020 year, which for the COVID-19 based studies should be explained. COVID-19 in China was observed since 2019, and even in Europe it was observed since 2020, Authors should be prepared to include the studies from all the countries. Taking this into account, even if they will not find any studies for 2019, they should conduct such searching.
(4) Authors used quite unusual measure to assess the quality of included studies, as they decided to indicate as follows: “Quality rating was performed according to criteria by Armijo-Olivo et al. [21].”. This description is quite unusual, as within the referred study no tool is presented. Authors of the referred study compare 2 tools, but they do not develop any novel one. Moreover, if Authors based on the referred study, they should follow the conclusions from it, as authors of the study indicate as follows: “The newly introduced CCRBT assigned these studies a higher risk of bias. Its psychometric properties need to be more thoroughly validated, in a range of research fields, to understand fully how to interpret results from its application.”. Based on the indicated conclusion of by Armijo-Olivo et al. [21] it must be stated, that we still do not know how to interpret the results obtained by using this tool.
(5) The major part of the manuscript are 2 tables (Table 1 and Table 2), which do not present the essential information. Authors should include to them also other important information from the included studies (e.g. time when the study was conducted, tools which were used to assess intake, inclusion and exclusion criteria, characteristics of the studied group, detailed observations, conclusions formulated by authors of the included studies, etc.), while they should preferably divide them into smaller tables. Similarly, the quality assessment should be presented in details within the table.
Author Response
Firstly, we would like to thank the Second Reviewer for the interest in our manuscript and the useful suggestions. We extensively revised the manuscript, according to your indications.
(1) The main problem with the presented study is the fact that Authors declare that their systematic review and meta-analysis “followed the Preferred Reporting Items for Systematic Reviews and Meta-Analyses (PRISMA) guidelines” (lines 112-114), but in fact the manuscript is not prepared according to the recommendations of PRISMA, which are very specific and should be rigorously followed. Authors should get familiar with PRISMA checklist (http://prisma-statement.org/prismastatement/Checklist.aspx) and they should correct their manuscript to be prepared according to the checklist. E.g., the Abstract should include: background, objectives, data sources, study eligibility criteria, participants, and interventions, study appraisal and synthesis methods, results, limitations, conclusions and implications of key findings, systematic review registration number, while a number of elements is not presented in the Abstract of the submitted manuscript. However, Authors should correct the whole study (not only the Abstract Section).
Thank you for your observations. We reviewed the PRISMA-abstract-checklist and modified the abstract accordingly, to ensure all items were included. We left a single paragraph without headings, according to the journal instructions. We also revised Methods (see details below) and Results, and we added a “Limitations” subparagraph to the Discussion. We added the PROSPERO registration number both in the Methods section and in the Abstract.
(2) Especially while analysing the Materials and methods Section, a majority of required information is not presented, so in fact the majority of essential information is not provided, so the quality of this systematic review can not be assessed.
Thank you for your comment. We extensively revised the Methods section, following the PRISMA checklist as you suggested.
(3) Authors conducted the systematic review based on the searching from 2020 year, which for the COVID-19 based studies should be explained. COVID-19 in China was observed since 2019, and even in Europe it was observed since 2020, Authors should be prepared to include the studies from all the countries. Taking this into account, even if they will not find any studies for 2019, they should conduct such searching.
Thank you for your observations. We realized that our wording was equivocal. Our search was actually conducted from 1987 to May 5th 2023; however, the search did not retrieve any article published before 2020. This is likely because, although COVID 19 had been observed since 2019, there was not enough time for studies to be completed before the end of 2019. We corrected our wording to avoid ambiguity.
(4) Authors used quite unusual measure to assess the quality of included studies, as they decided to indicate as follows: “Quality rating was performed according to criteria by Armijo-Olivo et al. [21].”. This description is quite unusual, as within the referred study no tool is presented. Authors of the referred study compare 2 tools, but they do not develop any novel one. Moreover, if Authors based on the referred study, they should follow the conclusions from it, as authors of the study indicate as follows: “The newly introduced CCRBT assigned these studies a higher risk of bias. Its psychometric properties need to be more thoroughly validated, in a range of research fields, to understand fully how to interpret results from its application.”. Based on the indicated conclusion of by Armijo-Olivo et al. [21] it must be stated, that we still do not know how to interpret the results obtained by using this tool.
Thank you for your observation. To assess the quality of included studies, we used the “Quality Assessment Tool for Quantitative Studies” developed by the Effective Public Health Practice Project (EPHPP). As indicated by Armijo-Olivo and colleagues (2012), the latter would appear to present better inter-rater agreement as compared to the Cochrane Collaboration Risk of Bias Tool. We changed our wording to avoid ambuiguity.
(5) The major part of the manuscript are 2 tables (Table 1 and Table 2), which do not present the essential information. Authors should include to them also other important information from the included studies (e.g. time when the study was conducted, tools which were used to assess intake, inclusion and exclusion criteria, characteristics of the studied group, detailed observations, conclusions formulated by authors of the included studies, etc.), while they should preferably divide them into smaller tables. Similarly, the quality assessment should be presented in details within the table.
Thank you for your suggestions. We expanded the tables, following your indications. Of note, information on the period of time during which each study was conducted has already been included (see “STUDY PERIOD” column). We specified this aspect in the Methods section “2.4 Data extraction” to avoid misunderstandings. We also added detailed information on quality assessment in the Methods section.
Round 2
Reviewer 2 Report
The manuscript entitled „ The Impact of the COVID-19 Pandemic on Binge Eating Disorder: a Systematic Review” presents interesting issue, but some problems should be corrected.
The major part of the manuscript are 2 tables (Table 1 and Table 2), which do not present the essential information. Authors should include to them also other important information from the included studies (e.g. time when the study was conducted, tools which were used to assess intake, inclusion and exclusion criteria, characteristics of the studied group, detailed observations, conclusions formulated by authors of the included studies, etc.), while they should preferably divide them into smaller tables. Similarly, the quality assessment should be presented in details within the table.
Author Response
First of all, we would like to thank the reviewer for the useful comments aimed to improve the present systematic review.
The major part of the manuscript are 2 tables (Table 1 and Table 2), which do not present the essential information. Authors should include to them also other important information from the included studies (e.g. time when the study was conducted, tools which were used to assess intake, inclusion and exclusion criteria, characteristics of the studied group, detailed observations, conclusions formulated by authors of the included studies, etc.), while they should preferably divide them into smaller tables. Similarly, the quality assessment should be presented in details within the table.
Thank you for the suggestion. We have already addressed your first request, adding to tables some important information that were missing. Moreover, in this new version of the manuscript, we divided the 2nd table into 2 smaller tables as you requested, and we detailed the quality assessment within the table.
The corrections have been incorporated and labelled in yellow in the current revision of the article. We appreciated your comments and we believe to have effectively addressed all the comments risen by you.